# Layer-by-Layer Pirfenidone/Cerium Oxide Nanocapsule Dressing Promotes Wound Repair and Prevents Scar Formation

**DOI:** 10.3390/molecules27061830

**Published:** 2022-03-11

**Authors:** Junwei He, Xinxian Meng, Chen Meng, Jiayu Zhao, Yunsheng Chen, Zheng Zhang, Yixin Zhang

**Affiliations:** 1Department of Plastic and Reconstructive Surgery, Shanghai Ninth People’s Hospital, School of Medicine, Shanghai Jiao Tong University, 639 Zhizaoju Rd., Shanghai 200011, China; hejunwei9623@sjtu.edu.cn (J.H.); msharon@sjtu.edu.cn (X.M.); 2College of Chemistry, Chemical Engineering and Biotechnology, Donghua University, Shanghai 201620, China; 2150554@mail.dhu.edu.cn (C.M.); 2200707@mail.dhu.edu.cn (J.Z.); 3Department of Burns and Plastic Surgery, Shanghai Institute of Burns Research, Ruijin Hospital Affiliated to School of Medicine, Shanghai Jiao Tong University, Shanghai 200025, China

**Keywords:** wound repair, scar prevention, cerium oxide nanoparticles, pirfenidone, nanocapsules, dressing

## Abstract

An increase in the levels of reactive oxygen species (ROS) and high expression levels of transforming growth factor-β (TGF-β) in wound tissue are two major problems for wound repair and scar inhibition. Modulation of the wound microenvironment is considered to be able to overcome these issues. Two possible solutions include the use of cerium oxide nanoparticles (CeO_2_) as an enzyme-like ROS scavenger and pirfenidone (PFD) as an anti-fibrotic drug to inhibit the expression of TGF-β. However, CeO_2_ is easily adsorbed by biological macromolecules and loses its enzyme-like activity. Furthermore, the intracellular delivery of PFD is difficult. Herein, the layer-by-layer method was used to prepare nanocapsules (NCs) with a sophisticated structure featuring PFD at their core and CeO_2_ in their shell; these NCs were referred to as PFD/CeO_2_ NCs. PFD/CeO_2_ NCs were supposed to efficiently achieve intracellular delivery of PFD and successfully scavenged ROS from the microenvironment. Cellular experiments verified that PFD/CeO_2_ NCs had good biocompatibility, satisfactory cellular uptake, and favorable ROS-scavenging capacity. To be applied directly to the wound, PFD/CeO_2_ NCs were then adhered to plasma-etched polylactic acid (PLA) fiber membranes to prepare a new wound dressing. Animal experiments further demonstrated that the dressing accelerated the epithelialization of the wound, reduced the levels of ROS and TGF-β, improved the arrangement and proportion of collagen fibers, and finally, achieved satisfactory wound-repairing and anti-scarring effects. These results provide a new concept for promoting wound repair and preventing scar formation.

## 1. Introduction

Human skin is limited by its own ability to repair itself following injuries that penetrate beyond the epidermis. Without appropriate medical treatment, the cutaneous wound-healing process mostly results in undesirable fibrosis and scar formation, in which an excessive amount of extracellular matrix (ECM) is deposited [1]. Even though scarring does restore integrity, it is not a necessary process and can be unsightly and detrimental to tissue function. Hence, wound repair and scar prevention remain the central concerns for dermatological clinical care [2]. Typical wound regeneration is a dynamic process in the microenvironment, involving a multitude of cells and biological signaling molecules. This coordinated process consists of three sequential and overlapping phases, including an inflammation phase, a proliferation phase, and a remodeling phase [3]. Most of the current treatments only focus on restoration of the missing structure, thus resulting in ill-suited healing and scarring [4]. Therefore, there is an urgent need to promote wound repair as well as to prevent scar formation through multipath microenvironment modulation.

The presence of reactive oxygen species (ROS) in the microenvironment is a double-edged sword for wound repair. As part of the defense mechanism against invading pathogens in the inflammation phase and redox messengers in the proliferation phase, ROS are produced by mitochondria, peroxisomes, cytochrome P-450, and many others [5]. If the wound becomes infected, respiratory burst of immune cells may lead to the overproduction of ROS. Excessive ROS can damage biological macromolecules (such as proteins, carbohydrates and DNA), thus causing a range of deleterious effects such as cellular senescence, inflammation and fibrotic scarring [6]. A practical way to promote wound repair by lowering the ROS level.

Therefore, ROS-modulating therapy is an emerging and promising candidate for promoting wound repair, with both natural and artificial ROS scavengers [7]. Physiologically, several protective enzyme systems work as ROS scavengers to maintain a redox balance and reduce ROS-induced oxidative stress damage, including superoxide dismutases (SOD), catalases (CAT) and glutathione peroxidases (GPx) [8]. Of all ROS scavengers, cerium oxide nanoparticles (CeO_2_) exhibit the most significant potential for various ROS-excessive diseases, such as tissue inflammation, acute organ injury, cancer, and chronic wounds [9]. The ROS-scavenging capacity of CeO_2_ originates from its large intrinsic reversible storage and release capacity for oxygen; this is associated with the formation of oxygen vacancy defects (OVDs) and the low potential redox transformation between Ce^3+^ and Ce^4+^ [10]. Thus, CeO_2_ is an effective enzyme-like ROS scavenger for the promotion of wound repair.

However, the nanoscale size and high superficial reactivity of CeO_2_ can lead to absorption with different proteins and peptides, forming a “protein corona” on the surface, thus limiting its cellular delivery and catalytic activity [11]. Nanocapsules are vesicular systems which constitute a drug core surrounded by polymeric membranes, with low toxicity and high loading capacity [12,13]. Recently, polyelectrolyte nanocapsules (NCs) have been regarded as feasible carriers for the delivery of various drugs, and metallic oxide nanoparticles (such as TiO_2_, SiO_2_ and Fe_3_O_4_) can be introduced into these NCs by applying the layer-by-layer (LbL) method [14]. In this form, CeO_2_ is usually stabilized by citric acid and is negatively charged in a water system. The positive charge of the cationic polyelectrolyte enables it to adhere to the anion polyelectrolyte layer and vice versa. Sandwiched by two positive-charged layers in the shell, CeO_2_ can be effectively incorporated into biodegradable NCs made with dextran sulfate sodium salt (DS) and poly-L-arginine hydrochloride (PArg) [15]. The shell of these NCs can be easily penetrated by ROS while shielding CeO_2_ from interference caused by additional proteins and peptides. In the form of NCs shell, CeO_2_ can maintain ROS-scavenging capacity both extracellularly and intracellularly, and provide a highly efficient option for promoting wound repair [16].

The major cause of scar formation in the microenvironment is the upregulation of transforming growth factor-beta (TGF-β). As a classical signaling pathway during the proliferation and remodeling phase, TGF-β is present at elevated levels and drives a shift in the phenotype of fibroblasts to myofibroblasts [17]. Myofibroblasts produce collagen and fibronectin in the ECM while downregulating matrix metalloproteinases to inhibit ECM turnover [15]. In cases of dysregulated healing, the excessive accumulation of ECM results in tissue fibrosis and even scar formation. Therefore, a feasible approach to inhibit scar formation is by reducing the expression of TGF [18].

One promising candidate for TGF-β modulation is pirfenidone (PFD), an FDA-approved broad-spectrum anti-fibrotic drug [19]. By reducing the over-expression of TGF-β, PFD lessens the differentiation of human fibroblasts into myofibroblasts and thus prevents the scarring process in several organs [20]. By targeting intracellular signaling pathways, PFD faces the same problem of inefficient cellular delivery, especially due to its small molecular weight (MW) and instability to ROS [21]. Fortunately, due to the hollow structure of NCs, the loading of PFD into NCs might be a possible solution to the cellular delivery problem. Regarding PFD, the CeO_2_ layer can work as a protectant against ROS and a co-agent for wound regeneration [22]. Therefore, NCs with a sophisticated structure featuring PFD at their core and CeO_2_ in their shell are assumed to help promote wound repair and prevent scar formation; these NCs are referred to as PFD/CeO_2_ NCs.

To be directly applied to a wound, an ideal wound dressing is a must for loading PFD/CeO_2_ NCs. Sponges, hydrogels, and fibers have emerged as promising materials for wound dressings; of these, electrospun fibers have attracted intense research attention [23]. Electrospinning is a simple and effective method that adjusts and controls the diameter, shape, and surface features of the fibers [24]. Electrospun-fiber membranes can maintain wound hydration, absorb excess wound exudate, and create a barrier for external microorganisms in a traditional manner. Moreover, these membranes can also mimic the ECM and create a positive environment for healing due to their superficial area and high porosity [25]. Polylactic acid (PLA) is the most widely used electrospun biopolymer due to its excellent biocompatibility and biodegradability. In addition, PLA fibers can be degraded into lactic acid by enzymes, thus acting as a pH regulator for wound regeneration [26]. Therefore, PLA-fiber membranes loaded with PFD/CeO_2_ NCs are an ideal wound dressing for promoting wound repair and preventing scar formation.

In this study, a robust intracellular delivery system with PFD and CeO_2_ in the form of polyelectrolyte NCs was developed. CeO_2_ was incorporated into the shell, and PFD was loaded as the core. The biodegradable PFD/CeO_2_ NCs provide controlled loading and intracellular release. A comprehensive analysis of their biocompatibility, cellular uptake, and ROS-scavenging capacity was carried out through cellular experiments. Then, the NCs were adhered on PLA-fiber membranes as an integrated wound dressing. The wound-healing and anti-scarring effect of the dressing was then determined in animal experiments, which included its influence on epithelialization, the levels of ROS and TGF-β, and the arrangement and proportion of collagen fibers in the wound. The PFD/CeO_2_-NC-loaded PLA dressing not only displayed the therapeutic functions of CeO_2_ and PFD, but also avoided the defects of their direct application. Collectively, those findings provide new insight into the design of composite functional wound dressings for promoting wound repair and preventing scar formation (Figure 1).

## 2. Experiential Section

### 2.1. Synthesis of PFD/CeO_2_ NCs

Aqueous colloid solutions of citric acid-stabilized CeO_2_ were synthesized as recorded [27]. In brief, citric acid (0.24 g, Sinopharm, Shanghai, China) was dissolved in cerium chloride solution (CeCl_3_, 25 mL, 0.05 M, Rhawn, Shanghai, China) and rapidly poured into ammonia water (100 mL, 3 M, Sinopharm, Shanghai, China) with stirring (600 rpm), and then was boiled for 2 h.

Capsules were prepared via the alternate deposition of oppositely charged polyelectrolytes on calcium carbonate (CaCO_3_) templates. Templates were synthesized by rapidly mixing equivalent solutions of calcium chloride (CaCl_2_, 0.1 M, Sinopharm, Shanghai, China) and sodium carbonate (Na_2_CO_3_, 0.1 M, Sinopharm, Shanghai, China). After intense agitation (2 min, 14,000 rpm) with a high-speed homogenizer (T18, IKA, Staufen, Germany), a suspension of submicron-sized templates was formed; this was then separated by centrifugation and rinsed three times.

The first polyelectrolyte layer was adsorbed on the surface of the template from positively charged PArg (70 kDa, 1 mg/mL, 0.15 M NaCl, Sigma-Aldrich, St Louis, MO, USA) solution by 1 h of incubation and agitation. The next layer was prepared by the absorption of negatively charged DS (10 kDa, 1 mg/mL, 0.15 M NaCl, Sigma-Aldrich, St Louis, MO, USA) solution. A solution of CeO_2_ was used to form the fourth layer. After each adsorption step, the capsules were thoroughly washed twice to remove uncoupled polymers. These processes were carried out alternately until the number of layers met our requirements (4/6/8 layers). CaCO_3_ templates were then etched by ethylene diamine tetra acetic acid solution (0.2 M, EDTA, Sinopharm, Shanghai, China) and pH was adjusted to 7. The resulting hollow capsules were centrifuged and washed twice.

Capsules were re-dispersed in a PFD solution (15 mg/mL, Aladdin, Shanghai, China) with stirring (600 rpm). After 1 h of incubation, the solution was heated to 80 °C for another hour before cooling and centrifugal ultrafiltration. The filtrate was collected to measure the loading rate and the resultant PFD/CeO_2_ NCs were washed and maintained in triple-distilled water.

To investigate the role of heating, some of the capsules were incubated at room temperature for 2 h in PFD solution. Furthermore, to investigate the effects of the two main components, PFD and CeO_2_, in subsequent experiments, PFD NCs and CeO_2_ NCs were prepared by replacing PFD solution with water during synthesis or replacing CeO_2_ with DS as the fourth layer, respectively.

### 2.2. Characterization of the NCs

The morphological features of CeO_2_ nanoparticles, CaCO_3_ templates, unheated hollow capsules, heated hollow NCs, unheated PFD-loaded capsules and heated PFD-loaded NCs were determined by transmission electron microscopy (TEM, 200 kV, JEM-2100F, JEOL, Tokyo, Japan). The zeta potential of CeO_2_ nanoparticles and each layer of capsules were measured during synthesis on a Zeta sizer analyzer (Nanotrac Wave II, Malvern, Malvern, UK) through electrophoretic light-scattering technique. Loading rate, embedding rate and the cumulative release of PFD after loading were calculated by using an UV−visible spectrophotometer (UV-vis, λ: 200–600 nm, UV-1800PC, Mapada, Shanghai, China). Loading rate referred to the ratio of PFD loss during the loading process. Embedding rate referred to the ratio of PFD inside NCs to total PFD in the resultant NC suspension.

### 2.3. Cell Culture

Normal human fibroblasts (NHF) were isolated, cultured and passaged within 2 h of post-surgical excision [28]. Cells from passage 3 to 6 were used in the following experiments. All tissues were collected from Shanghai 9th People’s Hospital and all patients provided written consent prior to surgery. The study was approved by the Ethics Committee of Shanghai Jiao Tong University School of Medicine, Shanghai, China. Three different types of NCs (PFD/CeO_2_ NCs, CeO_2_ NCs and PFD NCs) were tested in the following cell experiments.

### 2.4. CCK-8 Assay

The viability of NHFs incubated with NCs was assessed by performing cell counting kit-8 assays (CCK-8, Dojindo, Kumamoto, Japan). Twenty-four hours after NHFs were seeded, the medium was replaced with a medium containing NCs at five different concentrations. Cell viability was assessed after 24 h by a microplate reader (Synergy 2, Bio Tek, Winooski, VT, USA). The concentration with the highest viability (no less than 90% viability) was set as the “working concentration”. If not specified, the concentration of different NCs in the following experiments was the working concentration.

### 2.5. Cellular Uptake Assay

NC suspension was loaded with rhodamine B (RhB, TCI, Tokyo, Japan), and is referred to hereafter as RhB/CeO_2_ NCs and RhB NCs (prepared by replacing PFD with RhB). The equivalent RhB solution was used to evaluate nanocapsule internalization. Twenty-four hours after NHFs were seeded, the medium was replaced with a medium containing RhB NC suspension or solution. Microphotographs were taken with an inverted fluorescence microscope (DMi8, Leica, Bensheim, Germany) after 24 h of incubation. images were analyzed by image analysis software (Image-Pro Plus 6.0, Media Cybernetics, Bethesda, MD, USA).

### 2.6. Live/Dead Assay

Assessment of the viability of NHFs when cultured in the presence of different NCs was verified with a calcein-AM/PI double-staining kit (Live/dead Kit, Dojindo, Tokyo, Japan). Twenty-four hours after NHFs were seeded with NCs, a mixed-dye solution was added for 15 min. Images were taken on fluorescence microscope at an excitation wavelength of 490 nm (Live) and 545 nm (Dead), and measured by Image-Pro Plus 6.0.

### 2.7. ROS Assay

The levels of ROS in NHFs were determined using a reactive oxygen species assay kit (ROS assay kit, 2,7-dichlorodihydrofluorescein diacetate, DCFH-DA, S0033M, Beyotime, Shanghai, China). Non-fluorescent DCFH-DA could enter the cells freely and be oxidized by intracellular ROS into green-fluorescent 2,7-Dichlorofluorescein (DCF). The level of ROS then could be determined by the fluorescence of DCF. NHFs were pre-incubated with different NCs and hydrogen peroxide (H_2_O_2_, 300 μM, Aladdin, Shanghai, China) for 24 h. Then, ROS assay kit was added into medium. After 30 min, NHFs were washed with PBS, images were taken with a fluorescence microscope, and analysis was performed with Image-Pro Plus 6.0.

### 2.8. Scratch Assay

The migration of NHFs was determined using a monolayer wound assay in vitro. NHFs were seeded and plated until completely confluent. Then, they were scraped across the plate with a 200 μL pipette tip and cultured with low-serum Dulbecco’s modified eagle medium (DMEM, Gibco, Grand Island, NY, USA) with NCs for 48 h. Cell migration at 0, 24, and 48 h were imaged with an inverted microscope (BM-37XB, BM, Shanghai, China).

### 2.9. Synthesis and Characterization of Capsule-Loaded PLA Dressing

PLA (7%, Nature Works, Minnetonka, MN, USA) was dissolved in a mixed solvent of chloroform and acetone (3:1 by volume), heated to 60 °C for 1 h and fabricated into fibers by electrospinning (working distance: 15 cm, pumping rate: 1 mL/h, voltage: 18 kV). PLA fibers were collected on foil, treated with a plasma-etching machine, and punched into 6-millimeter-diameter-round dressings with a puncher.

The dressings were impregnated into different NC suspensions, left to dry, and preserved. The morphological characteristics of loaded/unloaded PLA dressings were observed with a scanning electron microscope (SEM, 20 kV, FlexSEM 100II, Hitachi, Tokyo, Japan).

### 2.10. Mouse Wound-Healing Model

Sprague Dawley mice (SD mice, 180–220 g, Shanghai Jessie Experimental Animal, Shanghai, China) were randomly divided into 5 groups (*n* > 5): (1) Control group: no treatment; (2) PLA group: treated with PLA dressing; (3) PFD NCs + PLA group: treated with PFD-NC-loaded PLA dressing; (4) CeO_2_ NCs + PLA group: treated with PFD-NC-loaded PLA dressing, and (5) PFD/CeO_2_ NCs + PLA group: treated with PFD/CeO_2_-NC-loaded PLA dressing. For each experimental animal, the fur on the dorsum was first shaved off. Then, four 6-millimeter-diameter full-thickness wounds were produced. Once the model had successfully been established, different dressings were applied to the mice in each group. All operations and euthanasia were performed under anesthesia. All experiments were approved and performed under the guidelines of the Ethics Committee of Shanghai Jiao Tong University School of Medicine.

### 2.11. Evaluation of Wound Closure

Digital images of the wounds were taken every 2 days until at least one group had healed completely. Wounds were analyzed by tracing the wound margins and performing calculations on Image-Pro Plus 6.0.

### 2.12. Histological Staining

Once the wound bed had completely re-epithelized in one of the groups, all mice were executed, and full-thickness, cross-sectional tissue samples were obtained for following analysis.

#### 2.12.1. ROS Assay

Specimens were prepared into frozen sections which were incubated with superoxide anion fluorescent probe (Dihydroethidium, DHE, S0063, Beyotime, Shanghai, China) at 37 °C for 30 min. Non-fluorescent DHE could be ingested by the cells and dehydrogenated by intracellular superoxide anions into ethidium, which emitted red fluorescence by binding to RNA or DNA. The sections were then examined by fluorescent microscopy.

#### 2.12.2. TGF-β Assay

Specimens were fixed in 4% paraformaldehyde (Biosharp, Hefei, China), embedded in paraffin wax and then sectioned. The sections were incubated with anti-TGF-β rabbit monoclonal antibody (1:50, Proteintech, Chicago, IL, USA) at 37 °C for 2 h, followed by incubation with horseradish peroxidase-conjugated goat anti-rabbit IgG (1:50, Proteintech, Chicago, IL, USA) for 30 min. The sections were then washed with PBS and treated with DAB detection kit (Proteintech, Chicago, IL, USA). Finally, sections were examined by microscopy.

#### 2.12.3. Collagen Assay

Specimens were fixed, embedded in paraffin, and sectioned. Masson’s trichrome staining was performed for epithelialization and total collagen analyses. For Sirius red staining, sections were immersed in Sirius Red (Servicebio, Wuhan, China) for 1 h at 37 °C. Under polarized light microscopy (DM2700P, Leica, Bensheim, Germany), collagen I (COL 1) fibers were stained orange, whereas collagen III (COL 3) fibers appeared green.

### 2.13. Statistical Analysis

Primary data were presented as means ± standard error of the mean and analyzed in Origin (version 2019b, OriginLab Corporation, Northampton, UK). Statistical analyses were performed using one-way ANOVA. The threshold of statistical significance is set to *p* < 0.05 and asterisks denote statistical significance (*, *p* < 0.05; **, *p* < 0.001; ***, *p* < 0.0001).

## 3. Results and Discussion

### 3.1. Preparation and Characterization of PFD/CeO_2_ NCs

Polyelectrolyte NCs were formed based on electrostatic interactions between polyelectrolytes with opposite charges. Biodegradable DS and PArg were used for most of the NC shells in this research due to their excellent dispersion and low cohesion [29].

Figure 1 shows the specific preparation route for PFD/CeO_2_ NCs. The synthesized CaCO_3_ templates were assembled alternately with positively charged PArg and negatively charged DS or CeO_2_. After forming the shell, the templates were dissolved with EDTA to obtain hollow capsules. After immersion in a high concentration of PFD solution for drug loading, the shell was sealed by heating. PFD was retained as the core in the NCs.

The CaCO_3_ templates could be made to contain core drugs by blending drugs with Na_2_CO_3_ or CaCl_2_ solution before preparation [30]. After removing the template with EDTA, only the core drugs were left in the capsules. In this research, by immersing the hollow capsules into a highly concentrated drug solution, the pores and osmotic pressure of the polyelectrolyte shell successfully delivered the drug to the core. Then, heating shrunk the shell and sealed the drug inside while reducing the volume of the capsule to the nanoscale. Essentially, during preparation, there were inorganic ions between the layers, in this case mainly Na^+^ and Cl^−^, thus creating external charge compensation [31]. With ions reducing the attraction between oppositely charged layers, the shells of the unheated capsule were left with gaps and the connection between layers remained incompact [32]. Heating promoted the return of ions into the native pairs outside, thereby supporting cross-linking between polymers and increasing internal charge compensation. This resulted in a denser, more structured hierarchical structure. The former method is suitable for biological macromolecules with high MW and poor thermal stability, while PFD has lower MW and better thermal stability and was more suited to the latter.

The size of the templates depended on the stirring speed in preparation. Most of the previous PArg/DS capsules were obtained at 600–800 rpm with a mean diameter of 2–3 μm [14,30]. To ensure successful cellular uptake, the mean diameter of the template reached 665.2 ± 16.8 nm in this research. After heating, the diameter of the NCs reached the nanometer magnitude, which made capsule internalization easier.

The zeta potential of citric acid-stabilized CeO_2_ nanoparticles was −18.6 ± 2.59 mV; this ensured the formation of a stable layer. TEM showed that the particle size was approximately 2–3 nm and that the particles were monodispersed with a good isotropic shape. The characterization of its other physicochemical properties was available in related references [27]. CeO_2_ exhibited high enzyme-like antioxidant activity, but could easily lose its enzyme-like properties by interacting with biomolecules. In previous studies, CeO_2_ was mostly placed in the fourth layer of the shell; this could not only prevent it from falling off during synthesis, but also prevent the formation of a “protein corona” [14]. Therefore, the CeO_2_ layers were prepared as the fourth layer in all the NCs containing CeO_2._

The number of layers was important as it determines the mechanical and chemical properties of NCs. Typically, each polyelectrolyte layer was approximately 5–6 nm [33]. The higher the number of layers, the stronger the mechanical structure and the more difficult the drug-loading. It has been reported that the volume of capsules with an even number of layers decreased while those with an odd number of layers increased after heating [32]. To increase the efficiency of intracellular delivery, drug loading rate, embedding rate and release rate of 4/6/8-layer heated NCs were first investigated (Figure 1A,B). There was no significant difference in loading rates, which were all around 20–30%, and the embedding rate of 6-layer NCs was significantly higher than the other two. This indicated that PFD was most preserved in the 6-layer NCs. Furthermore, the PFD release rate from 6-layer NCs was the lowest and gentlest. With 4 layers, the shell might not be of a sufficient thickness to prevent PFD from easily escaping. With 8 layers, most of the PFD might fail to enter the core and remain in the shells, thus leading to uncontrolled release. Therefore, the 6-layer structure was optimal. The performance of heated and unheated 6-layer capsules was also compared, and it was found that the heated capsules could block PFD better, thus ensuring efficient intracellular delivery.

Zeta potential was measured during synthesis to ensure each layer was assembled completely (Figure 1C). Due to its crystal structure, CaCO_3_ templates were not an ideal regular sphere and were relatively easy to polymerize into 3–5 μm masses with their small particle size (Figure 1D). As the outer polyelectrolyte layer was increased, the dispersion improved; this was due to charge repulsion. After assembly, the polyelectrolyte shell was visible on the template when examined by TEM (Figure 1E). By removing CaCO_3_ with EDTA, the size of the capsule was reduced to approximately 419.8 ± 14.7 nm (Figure 1F and Figure 2A). The diameter of the hollow NCs was significantly lowered to approximately 72.6 ± 4.15 nm after heating (Figure 1G and Figure 2B). Following immersion in PFD solution, the internal water seeped out under osmotic pressure, and capsules shrunk to approximately 211.3 ± 8.9 nm (Figure 1H and Figure 2C), while the size of the loaded NCs was approximately 111.1 ± 4.7 nm after heating (Figure 1I and Figure 2D). This difference in volume indirectly indicated that PFD had been successfully enclosed in the capsules. Furthermore, the final size of the NCs met the requirements for intracellular uptake. It was observed that each type of capsule was evenly distributed with small black spots, thus indicating that the CeO_2_ layer had been evenly assembled.

### 3.2. CCK-8 Assay

The working concentration of the prepared NC suspensions was determined for use in subsequent experiments. A CCK-8 kit was used to observe cell viability in the medium containing 1, 5, 10, 15 and 20% of NC suspension (Figure 3A). With an increase in concentration, the cell viability showed a downwards trend, although viability remained over 50%. The viability was highest in PFD NCs group, followed by CeO_2_ NCs group and PFD/CeO_2_ NCs group; this trend was due to the subtle toxicity of CeO_2_. The working concentration, in this case, was 15%.

### 3.3. Cellular Uptake Assay

To verify that the NCs had been taken up by cells, PFD was replaced with fluorescent RhB as the core drug. RhB has high levels of thermal stability and low cytotoxicity with a MW of the same magnitude as PFD. In cells, RhB exhibited obvious red fluorescence under fluorescence microscopy; the higher the fluorescence intensity, the higher the cellular uptake (Figure 3B). There was no prominent difference in cellular fluorescence when compared between the RhB/CeO_2_ NCs group and the RhB NCs group, although the fluorescence in the RhB-solution group was relatively lower (Figure 3C). This indicated that the intracellular delivery efficiency of NCs was higher than that of aqueous solution, and that the presence or absence of the CeO_2_ layer had no apparent effect on cellular uptake.

### 3.4. Live/Dead Assay

The biosafety of the NCs at the determined working concentration was verified by application of a Live/dead kit. Under a fluorescence microscope, living cells appeared green while dead cells appeared red (Figure 3D). In accordance with the CCK-8 assay, the PFD NCs group, CeO_2_ NCs group, and PFD/CeO_2_ NCs group showed decreasing levels of viability, although viability in each case exceeded 90% (Figure 3E). The differences between the Live/dead assay and the CCK-8 assay may be related to systematic error of methods. 

### 3.5. ROS Assay

CeO_2_ can degrade ROS into H_2_O and O_2_ as an enzyme-like ROS scavenger (Figure 4A). The application of H_2_O_2_ at a concentration of 300 mM did not induce apoptosis, although the levels of ROS inside the cells increased significantly, thus simulating the inflammatory stage in wounds. NCs were co-cultured with NHFs for 24 h, and the intracellular levels of ROS were determined with a ROS assay kit (Figure 4B). Data showed that CeO_2_ NCs and PFD/CeO_2_ NCs significantly reduced the levels of ROS; this was due to the antioxidant ability of CeO_2_ (Figure 4C). It should be noted that the CeO_2_ layer played a key role in anti-ROS effects during intracellular delivery. However, PFD NCs had no antioxidant effect and were not significantly different from the control.

### 3.6. Scratch Assay

PFD NCs, CeO_2_ NCs and PFD/CeO_2_ NCs had no noteworthy inhibitory effect on the migration of NHFs compared with the control (Figure 4D).

### 3.7. Synthesis and Characterization of Capsule-Loaded PLA Dressing

A PLA-fiber dressing was prepared by electrospinning; the diameter of this fiber was 0.8–1 μm (Figure 5A). However, due to the hydrophobicity of the PLA surface, the NC suspension was not retained on the dressing throughout impregnation. Therefore, plasma-etching technology was used. This produced numerous pores and polar groups on the surface, thus forming van der Waals forces with NCs. Thus, the treated fibers were easily covered with NCs. After drying samples for SEM, it was found that some NCs easily polymerized into groups and became trapped between fibers (Figure 5A). If kept moist, capsule-loaded PLA dressings were equably coated with NCs.

### 3.8. Evaluation of Wound Closure

Different dressings were placed on the wounds created on the dorsum of experimental mice in each group (Figure 5B). On average, these dressings were changed every 2–3 days. The wound area was recorded during each dressing change. The wounds treated with PFD/CeO_2_ NCs + PLA group completely healed within 14 days, while most of the other groups remained unclosed; this was set as the cut-off point for experimental observation (Figure 5C).

According to statistical analysis, all groups showed a decreasing trend at the 14-day timepoint. In the early stages of wound repair, the PFD/CeO_2_ NCs + PLA group and CeO_2_ NCs + PLA group exhibited an obvious healing advantage. One week later, the gap between PFD NCs + PLA group and control group had significantly widened (Figure 5D). It was considered that during the early stages, the levels of ROS in the wound microenvironment were extremely high. This form of ROS was degraded by the CeO_2_, thus preventing cell injury and apoptosis due to excessive oxidative stress [34]. Subsequently, wounds tended to be more stable and the internalized PFD-loaded NCs disintegrated under osmotic pressure, thus releasing PFD and regulating tissue repair through the TGF-β pathway. Thus, the PFD NCs + PLA group had obvious healing superiority in the later stages. These data also proved that PLA itself could promote wound repair by regulating the pH of the wound and by inhibiting foreign bacteria.

### 3.9. Histological Staining

Frozen sections were used to detectsuperoxide anions by DHE probe. Since the main forms of ROS are superoxide anions, the number of superoxide anions in the section reflects the ROS levels. It was found that the levels of ROS in the PFD/CeO_2_ NCs +PLA group and CeO_2_ NCs + PLA group were lower than those in the other groups, thus indicating that the CeO_2_ layer played an enzyme-like role in ROS scavenging, thus improving early wound healing (Figure 6A,C).

Paraffin sections were immunohistochemically stained to detect TGF-β. TGF-β is a known inducer of fibroblast differentiation. Furthermore, the TGF-β/Smad signaling pathway is known to play a key role in promoting collagen synthesis and deposition in wound repair [17]. The expression levels of TGF-β in the PFD/CeO_2_ NCs +PLA group and PFD NCs + PLA group were significantly lower than in the other groups, indicating that PFD entered NHFs through dressings and successfully inhibited the expression of TGF-β, thus reducing the excessive synthesis of ECM and preventing scar formation (Figure 6B,C).

Masson staining (Figure 6D) and Sirius red staining (Figure 6E) further revealed collagen synthesis in the wounds. The total collagen content of the PFD/CeO_2_ NCs + PLA group, CeO_2_ NCs + PLA group, PFD NCs + PLA group, PLA group and control group showed a decreasing trend. However, collagen fibers became coarser and more irregular in the CeO_2_ NCs + PLA group than those in the PFD/CeO_2_ NCs + PLA group and PFD NCs + PLA group. The ratio of collagen I/III was the lowest in the PFD/CeO_2_ NCs + PLA group, followed by the PFD NCs + PLA group, CeO_2_ NCs + PLA group, PLA group, and control group. Although wound epithelialization in the CeO_2_ NCs + PLA group was faster than in the PFD NCs + PLA group, collagens were slightly more extensive, with a higher collagen I/III ratio (Figure 6F). Therefore, PFD played a significant role in collagen degradation and reconstruction; these are two key processes that underlie scar inhibition.

## 4. Conclusions

In this research, we engineered polyelectrolyte NCs containing PFD and CeO_2_. PFD was loaded as a core drug and CeO_2_ was combined into a capsule shell, thus ensuring effective intracellular PFD delivery and ROS scavenging. NHF experiments verified that PFD/CeO_2_ NCs had good levels of biocompatibility, satisfactory cellular uptake, and favorable ROS-scavenging capacity. Next, NCs were impregnated and adhered to PLA-fiber membranes, thus forming a new type of wound dressing. Experiments in mice showed that this type of dressing accelerated epithelialization of wounds, reduced the levels of ROS and TGF-β, improved the arrangement and proportion of collagen fibers, and finally, achieved a satisfactory wound-repairing and anti-scarring effect. Collectively, our findings provide a new concept for promoting wound repair and preventing scar formation.

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
