# Peer review of "Layer-by-Layer Pirfenidone/Cerium Oxide Nanocapsule Dressing Promotes Wound Repair and Prevents Scar Formation"

_molecules, 2022, doi:10.3390/molecules27061830_

Round 1
Reviewer 1 Report
The article reveals quite fully the effect of nanocapsules based on cerium oxide and an antifibrotic drug on wound healing and shows their effectiveness. However, there are a number of remarks and remarks to which the authors should pay attention.
In the review of the literature, the topic of the effect of ROS on wound healing is not sufficiently disclosed, it is necessary to highlight the positive role of ROS in this process and show under what conditions it becomes necessary to reduce their concentration.
Figure 3D does not show what the caption to the figure describes. It's probably a scratch test.
The method for indicating reactive oxygen species is not sufficiently well and clearly described. Figure 3B shows micrographs of cells with green fluorescence, while Figure 5A shows fluorescent micrographs of cells in the red spectrum. In general, in the materials and methods section, it is necessary to describe in more detail the method used for the identification of ROS, to describe in general terms what the kit for the determination of ROS is based on. In addition, data on the inhibition of ROS concentration under the influence of nanocapsules should be supplemented by the determination of another type of ROS, for example, superoxide. In this case, it would be more informative to provide data on the level of expression of antioxidant defense genes in cells.
Author Response
Response to reviewer #1
The article reveals quite fully the effect of nanocapsules based on cerium oxide and an antifibrotic drug on wound healing and shows their effectiveness. However, there are a number of remarks and remarks to which the authors should pay attention.
Comment 1: In the review of the literature, the topic of the effect of ROS on wound healing is not sufficiently disclosed, it is necessary to highlight the positive role of ROS in this process and show under what conditions it becomes necessary to reduce their concentration.
Response: Thank you for your suggestion. The content of this part has been modified. In fact, ROS is a double-edged sword for wound repair. ROS is part of the defense mechanism against invading pathogens in the inflammation phase and redox messengers in the proliferation phase. If the wound gets severely infected, respiratory burst of immune cells may lead to the overproduction of ROS. However, excessive ROS can damage biological macromolecules, thus causing a range of deleterious effects .In this case, it becomes necessary to reduce their concentration.
Comment 2: Figure 3D does not show what the caption to the figure describes. It's probably a scratch test.
Response: Thank you for your suggestion. That is a typing mistake, and it’s a scratch test. Now, the figure describes have been fixed.
Comment 3: The method for indicating reactive oxygen species is not sufficiently well and clearly described. Figure 3B shows micrographs of cells with green fluorescence, while Figure 5A shows fluorescent micrographs of cells in the red spectrum. In general, in the materials and methods section, it is necessary to describe in more detail the method used for the identification of ROS, to describe in general terms what the kit for the determination of ROS is based on. In addition, data on the inhibition of ROS concentration under the influence of nanocapsules should be supplemented by the determination of another type of ROS, for example, superoxide. In this case, it would be more informative to provide data on the level of expression of antioxidant defense genes in cells.
Response: Thank you for your suggestion. The description of the methods for indicating ROS has been added. It’s my fault not to point out the difference between cell and frozen sections. 2,7-dichlorodihydrofluorescein diacetate (DCFH-DA) is used to determine intercellular ROS in cell experiment, while dihydroethidium (DHE) is used to determine superoxide anions on frozen sections. Therefore, Figure 3B shows micrographs of cells with green fluorescence, while Figure 5A shows fluorescent micrographs of cells in the red spectrum.
The level of expression of antioxidant defense genes in cells can only provide circumstantial evidence of ROS inhibition. With direct evidences of like DCFH-DA staining in cells and DHE staining on the sections, determining the expression of antioxidant defense genes does not worth the effort. However, we may investigate it in the further research.
Reviewer 2 Report
Comments and Suggestions for Authors
The manuscript entitled “Layer-by-layer Pirfenidone/Cerium oxide Nanocapsule Dressing Promotes Wound Repair and Prevents Scar Formation” seems to be interesting. Please consider reviewing based on the suggestions below, in order to improve the quality of the article.
1.Introduction:
-The introduction section must be completed with definition of nanocapsules and what are the advantages over other nanosystems. I suggest several references concerning the nanocapsules: i) Raţă, D. M., Chailan, J.-F., Peptu, C. A., Costuleanu, M., & Popa, M. (2015). Chitosan: poly(N-vinylpyrrolidone-alt-itaconic anhydride) nanocapsules—a promising alternative for the lung cancer treatment. Journal of Nanoparticle Research, 17(7). doi:10.1007/s11051-015-3115-1; ii) Raţă, D. M., Popa, M., Chailan, J.-F., Zamfir, C. L., & Peptu, C. A. (2014). Biomaterial properties evaluation of poly(vinyl acetate-alt-maleic anhydride)/chitosan nanocapsules. Journal of Nanoparticle Research, 16(8). doi:10.1007/s11051-014-2569-x.
- Please better highlight the novelty of this study.
- Experiential section
- Please explain how the potential zeta analysis was performed.
- The stability of nanocapsules in aqueous solutions is very important for wound repair. Why the authors did not determine the stability of obtained nanocapsules in solutions that mimic wound exudate (pH; temperature)?
- A granulometric analysis of the nanocapsules is recommended.
- It is also recommended to perform a structural analysis (for example FTIR) of the obtained nanocapsules.
Author Response
Response to reviewer #2
Comment 1: The introduction section must be completed with definition of nanocapsules and what are the advantages over other nanosystems. I suggest several references concerning the nanocapsules: i) Raţă, D. M., Chailan, J.-F., Peptu, C. A., Costuleanu, M., & Popa, M. (2015). Chitosan: poly(N-vinylpyrrolidone-alt-itaconic anhydride) nanocapsules—a promising alternative for the lung cancer treatment. Journal of Nanoparticle Research, 17(7). doi:10.1007/s11051-015-3115-1; ii) Raţă, D. M., Popa, M., Chailan, J.-F., Zamfir, C. L., & Peptu, C. A. (2014). Biomaterial properties evaluation of poly(vinyl acetate-alt-maleic anhydride)/chitosan nanocapsules. Journal of Nanoparticle Research, 16(8). doi:10.1007/s11051-014-2569-x.
Response: Thank you for your suggestion. Now, the definition and the advantages of nanocapsules have been added and the references are cited as Ref. 12 and 13.
Comment 2: Please better highlight the novelty of this study
Response: Thank you for your suggestion. The main novelty of this study is that the PFD/CeO2-NC-loaded PLA dressing can not only display the therapeutic functions of CeO2 and PFD, but also avoid the defects of their direct application. This point is now highlighted in the introduction.
Comment 3: Please explain how the potential zeta analysis was performed.
Response: Thank you for your suggestion. Potential zeta analysis was performed with a Zeta sizer analyzer through electrophoretic light scattering (ELS) technique. We just sent the nanocapsules suspensions into the analyzer and the data of zeta potential are tested immediately. ELS is a technique used to measure the electrophoretic mobility of particles in dispersion. This mobility is often converted to Zeta potential to enable comparison of materials under different experimental conditions. The fundamental physical principle is that of electrophoresis. A dispersion is introduced into a cell containing two electrodes. An electrical field is applied to the electrodes, and particles that have a net charge, or more strictly a net zeta potential will migrate towards the oppositely charged electrode with a velocity, known as the mobility, that is related to their zeta potential.
Comment 4: The stability of nanocapsules in aqueous solutions is very important for wound repair. Why the authors did not determine the stability of obtained nanocapsules in solutions that mimic wound exudate (pH; temperature)?
Response: Thank you for your suggestion. Actually, the TEM images of heated nanocapsules were taken 2 weeks after preparation, because the TEM failed after we sended the samples and the repair took 2 weeks. In this case, the nanocapsules are kept in distilled water for 2 weeks. However, we didn’t observe any evidence of breaking nanocapsules. Therefore, we assumed the nanocapsules were sable enough for the application. We will definitely investigate it in the further research.
Comment 5: A granulometric analysis of the nanocapsules is recommended.
Response: Thank you for your suggestion. The granulometric analysis of the nanocapsules has been added in Figure 2.
Comment 6: It is also recommended to perform a structural analysis (for example FTIR) of the obtained nanocapsules.
Response: Thank you for your suggestion. The preparation of nanocapsules didn’t include the formation and breaking of the chemical bonds, only with the electrostatic interaction of PArg, DS and CeO2. All main components of nanocapsules were intact. Therefore, FTIR and other structural analysis seem not worth the effort. We might investigate it in the further research.